# Susceptibility to Bacteriocins in Biofilm-Forming, Variable Staphylococci Isolated from Local Slovak Ewes’ Milk Lump Cheeses

**DOI:** 10.3390/foods9091335

**Published:** 2020-09-22

**Authors:** Andrea Lauková, Monika Pogány Simonová, Valentína Focková, Miroslav Kološta, Martin Tomáška, Emília Dvorožňáková

**Affiliations:** 1Institute of Animal Physiology, Centre of Biosciences of the Slovak Academy of Sciences, Šoltésovej 4–6, 040 01 Košice, Slovakia; simonova@saske.sk (M.P.S.); fockova@saske.sk (V.F.); 2Dairy Research Institute, a.s. Dlhá 95, 010 01 Žilina, Slovakia; kolosta@vumza.sk (M.K.); tomaska@vumza.sk (M.T.); 3Parasitological Institute of the Slovak Academy of Sciences, Hlinkova 3, 040 01 Košice, Slovakia; dvoroz@saske.sk

**Keywords:** bacteriocins, inhibition, ewes’ milk cheese, staphylococci

## Abstract

Seventeen staphylococci isolated from 54 Slovak local lump cheeses made from ewes’ milk were taxonomically allotted to five species and three clusters/groups involving the following species: *Staphylococcus aureus* (5 strains), *Staphylococcus xylosus* (3 strains), *Staphylococcus equorum* (one strain) *Staphylococcus succinus* (5 strains) and *Staphylococcus simulans* (3 strains). Five different species were determined. The aim of the study follows two lines: basic research in connection with staphylococci, and further possible application of the bacteriocins. Identified staphylococci were mostly susceptible to antibiotics (10 out of 14 antibiotics). Strains showed γ-hemolysis (meaning they did not form hemolysis) except for *S. aureus* SAOS1/1 strain, which formed β-hemolysis. *S. aureus* SAOS1/1 strain was also DNase positive as did *S. aureus* SAOS5/2 and SAOS51/3. The other staphylococci were DNase negative. *S. aureus* SAOS1/1 and SAOS51/3 showed biofilm formation on Congo red agar. However, using quantitative plate assay, 12 strains out of 17 showed low-grade biofilm formation (0.1 ≤ A_570_ < 1), while five strains did not form biofilm (A_570_ < 0.1). The growth of all strains, including those strains resistant to enterocins, was inhibited by nisin and gallidermin, with high inhibition activity resulting in the inhibition zone in size from 1600 up to 102,400 AU/mL (arbitrary unit per milliliter). This study contributes to microbiota colonization associated with raw ewe’s milk lump cheeses; it also indicates bacteriocin treatment benefit, which can be used in prevention and/or elimination of staphylococci.

## 1. Introduction

Milk from farm animals is a good source of nutrients. Consumers increasingly demand traditional/local products, to which those made from ewes’ milk belong. Ewes’ milk is rich in dry matter, minerals (calcium, zinc, and phosphorus) and B group vitamins. The principal protein in ewes’ milk, casein, is tolerated by people who are allergic, for example to cows’ milk [1]. Products made from ewes’ milk are even more suitable for persons suffering from lactose intolerance. In ewes’ milk lump cheese, lactose is mostly removed into the whey during processing, which means that consumption of such cheese should be safe for lactose-intolerant people. Despite modernization and centralization of the dairy industry, many European cheeses are produced traditionally from raw or heat-treated milk with or without the use of natural undefined starters in farms [2]. This is also the case of local Slovak ewes’ milk lump cheese. It is a traditional Slovak cheese produced from unpasteurized ewes’ milk without any added culture. However, the microbiota in ewes’ milk comes from various sources. Of course, the predominant microorganisms are beneficial lactic acid bacteria [3]. The ratio of beneficial to other bacteria in milk can depend on various factors, such as the environment itself or sheepfold hygiene. It is necessary therefore to maintain suitable safety conditions during processing. Ewe’s milk lump cheese should contain at least 47% dry matter and 52% fat in dry matter [1]. Processing of ewe’s milk consists of the following phases: making cheese curd, treating of raw material, lump forming, and dripping [4]. During the application of rennet, the milk temperature is optimized according to the season between 28 °C and 30 °C. When the raw material is ready after rennet addition, it is turned and cut, and then it is left to settle. All these processes last for 15 min. The settled raw material is then put in a form and the whey is allowed to drain out. Each lump is left dripping at a temperature of 18–20 °C for 10–24 h and is then placed on a shelf at 13–15 °C with sufficient ventilation. A slight skin covering forms on the cheese during the course of three days, and small grain-sized holes develop inside the cheese; the taste of the final cheese is slightly acidic [4].

As previously mentioned, in spite of efforts to maintain hygiene conditions, bacteria such as staphylococci can contaminate ewes’ milk. Staphylococci belong in the phylum Firmicutes, the order Coccus, the family Staphylococcaceae, and the genus *Staphylococcus*. The taxonomy is based on 16S rRNA sequences and most of the staphylococcal species fall into 11 clusters [5]. Staphylococci belong also in a group of bacteria, which are able to form biofilm [6]. Biofilms are an aggregation of microorganisms attached to and growing on a surface, and any type of bacteria with the ability to form biofilm can play a key role in contamination and infection [7]. However, some antimicrobial substances such as bacteriocins can inhibit those biofilm-forming bacteria. Bacteriocins are antimicrobial proteinaceous substances with inhibition activity against more or less relative bacteria [8,9]. A special group among them consists of enterocins produced mostly by enterococci [8,9] and also lantibiotic bacteriocins such as nisin and gallidermin [10]. While enterocins are known to inhibit Gram-positive and Gram-negative bacteria under in vitro and in vivo conditions [11,12], nisin and gallidermin act mostly against Gram-positive bacteria [10], although under in vivo conditions their inhibition effect against Gram-negative species has also been reported [13,14]. To know occurrence of microbiota which can influence character and consumption of products is beneficial for both consumers and producers. However, considering that only limited information exists regarding the characterization of bacteria detected in ewes’ lump cheeses, this study was focused on staphylococci in local Slovak ewes’ milk lump cheeses. Isolated staphylococci were tested for susceptibility to antimicrobial substances-enterocins as well as to two lantibiotic bacteriocins, such as nisin and gallidermin. The aim follows two lines: basic research in connection with staphylococci in ewes’ lump cheese, and further possible utilization of bacteriocins to protect cheese.

## 2. Materials and Methods

### 2.1. Sample Collection, Isolation, and Identification of Staphylococci

A total of fifty-four fresh ewes’ milk lump cheeses from various local farm producers in central Slovakia were sampled at farms and transported in our laboratory. Samples were treated using the standard microbiological method specified by the International Organization for Standardization (ISO). They were mixed in peptone water (Merck, Darmstadt, Germany) using Stomacher (Masticator, IUL, Madrid, Spain) and decimal dilutions in Ringer solution were prepared (ratio 1:9, Merck, Germany). Diluted samples were spread onto selective media, Baird-Parker agar supplemented with egg yolk and tellurite, and on Mannitol Salt agar (MSA; Becton and Dickinson, Sparks, MA, USA), as recommended with ISO and cultivated at 37 °C for 24–48 h. Total bacterial count was expressed in log 10 colony forming unit per gram (log 10 CFU/g). Different colonies (80) were picked up, checked for purity and submitted for identification using the matrix–assisted laser desorption ionization time-of-flight spectrometry identification system (MALDI-TOF MS, Brucker Daltonics, Billerica, MA, USA) based on protein “fingerprints“ [15]. Lysates of bacterial cells were prepared according to the producer’s instructions (Bruker Daltonics, Billerica, MA, USA). The results were evaluated using the MALDI Biotyper 3.0 (Bruker Daltonics, Billerica, MA, USA) identification database. Taxonomic allocation was evaluated on the basis of highly probable species identification (value score 2300–3000), secure genus identification and/or probable species identification (2000–2299) and probable genus identification (1700–1999). Positive controls were those included in the Bruker Daltonics database. Colonies with the same MALDI-TOF score value were excluded. For subsequent testing, identified strains were maintained on MSA agar or Baird-Parker agar (Becton and Dickinson, Sparks, Maryland, USA) and stored using the Microbank system (Pro-Lab Diagnostic, Richmond, VA, Canada).

### 2.2. API Staph Profile-Identity Strip System

Identification was also checked with API STAPH strips, which are miniaturized biochemical tests with a specially adapted database. The strips contain dehydrated test substrates in individual microtubes. Each microtube contains a tested medium and isolate inoculated in API STAPH medium added to it. After 18–24 h incubation at 37 °C results are read and interpreted with reference to the information contained in the manual. Control strains are those involved in the test manual or Bergey‘s Manual of Systematic Bacteriology (*Staphylococcus xylosus* ATCC 35663; *S. simulans* ATCC 27848; American Type Culture Collection, Rockville, MA, USA). The tests involved are as follows: acidification due to carbohydrate utilization (glucose, fructose, mannose, maltose, lactose, trehalose, mannitol, xylitol, melibiose, raffinose, xylose, sucrose, α-methyl-D-glucoside, and N-acetyl-glucosamine), reduction of nitrate to nitrite, alkaline phosphatase, acetyl-methyl-carbinol production, arginine dihydrolase, and urease.

### 2.3. Antimicrobial Phenotype Testing, Determination of Nuclease, and Hemolysis Activity 

To determine the antimicrobial profile of the identified staphylococci, the CLSI system was applied using 14 antibiotic disks [16]. Antibiotic disks were used according to the suppliers‘ recommendation: oxacillin (1 µg), clindamycin (2 µg), lincomycin (2 µg), penicillin (10 IU), tobramycin (10 µg), neomycin (10 µg), ampicillin (10 µg), erythromycin (15 µg), azithromycin (15 µg), streptomycin (25 µg), chloramphenicol (30 µg), rifampicin (30 µg), vancomycin (30 µg), and cefoxitin (30 µg) supplied by Oxoid (ON, Canada). Overnight cultures (100 µL) of the tested strains were spread on Mueller-Hinton agar (Difco, Lawrence, KS, USA) and the appropriate disks were applied. Plates were cultivated at 37 °C for 24 h. *Staphylococcus aureus* CB44 (CCM, Brno, Czech Republic) served as positive control.

To determine nuclease activity, each strain was inoculated onto the surface of DNase agar (Oxoid, USA) and incubated for 24 h at 37 °C. After flooding and acidifying the medium with 1 N HCl, the DNA precipitated out; and the medium became turbid with clear zones around DNase-positive colonies. *S. aureus* SA4 from a dog (isolated in our laboratory by Dr. Strompfová) was used as positive control. 

Hemolysis was detected by streaking the cultures onto BH agar (Difco, Lawrence, KS, USA) supplemented with 5% defibrinated sheep blood (from our stall). Plates were incubated at 37 °C for 24–48 h in an incubator. Presence or absence of clear zones around the colonies was interpreted as α- and β-hemolysis, while those testing negative exhibited γ-hemolysis [17].

### 2.4. Virulence Profile (Slime Production)

Slime production or biofilm formation belongs to the virulence factors. This ability in the identified staphylococci was tested using two methods, the qualitative phenotypic method, and the quantitative method. Congo red agar is the medium used in the phenotypic qualitative method [18]. This medium was composed of brain–heart infusion (BHI, Difco; Detroit, Michigan, USA, 37 g/L) enriched with sucrose (36 g/L, Slavus, a.s. Bratislava, Slovakia), pure agar (30 g/L, Becton and Dickinson, Sparks, MA, USA) and Congo red dye (0.8 g/L, Merck, Germany). Plates of the medium were inoculated with the tested strains and incubated at 37 °C for 24 h. A positive result was indicated by black colonies with dry crystalline consistency. Non-slime producers usually remained pink. The color was also checked after 48 and 72 h. Each strain was tested in duplicate.

Biofilm formation measured by means of quantitative biofilm plate assay is a method focusing on the biofilm formation capacity of tested strains. One colony of the tested strain grown on brain–heart agar (BHA) overnight at 37 °C (Difco, Lawrence, KS, USA) was transferred into 5 mL Ringer solution (pH 7.0, 0.75% *w*/*v*) to reach McFarland Standard no. 1 corresponding to 1.0 × 10^8^ CFU/mL. A volume of 100 µL from that culture was then transferred into 10 mL of brain–heart infusion (BHI). Standardized culture (200 µL) was inoculated into polystyrene microtiter plate wells (Greiner ELISA 12 Well Strips, 350 µL, flat bottom, Frickenhausen GmbH, Germany) and incubated for 24 h at 37 °C. The biofilm formed in the microtiter plate wells was washed twice with 200 µL of deionized water and dried at room temperature for 40 min. The remaining attached bacteria were stained for 30 min at room temperature with 200 µL of 0.1% (*m*/*v*) crystal violet in deionized water. The dye solution was aspirated away, and the wells were washed twice with 200 µL of deionized water and dried at room temperature for 30 min. After water removal and drying, the dye bound to the adherent biofilm was extracted with 200 µL 95% ethanol and stirred. A 150 µL aliquot was transferred from each well and placed in a new microplate well for absorbance (A) measurement at 570 nm using an Apollo 11 Absorbance Microplate reader LB 913 (Berthold Technologies, Oak Ridge, TN, USA). Each strain and condition was tested in two independent tests with 12 replicates. A sterile culture medium (BHI) was included in each analysis as negative control. *Streptococcus equi* subsp. *zooepidemicus* CCM 7316 was used as positive control (kindly provided by Dr. Eva Styková, University of Veterinary Medicine and Pharmacy in Košice, Slovakia). Biofilm formation was classified as highly positive (A_570_ ≥ 1), low-grade positive (0.1 ≤ A_570_ < 1) or negative (A_570_ < 0.1) according to Chaieb et al. [19] and Slížová et al. [20].

### 2.5. Susceptibility to Bacteriocins 

Four enterocins (10 µL from each) characterized in our laboratory were used: enterocin M produced by *Enterococcus faecium* AL41 (CCM 8558), enterocin A/P produced by *Enterococcus faecium* EK13 = CCM7419, both of environmental origin [21,22], Ent 412 produced by *Enterococcus faecium* EF412, horse strain [23] and durancin ED26E/7 produced by food origin strain *Enterococcus durans* ED26E/7 [24]. Partially purified enterocins (precipitates) were used as previously reported [13,14,21,22,23]. Nisin and gallidermin used were commercial pure substances. Nisin (product Nisaplin) was supplied by Aplin and Barrett Ltd., Dorset, United Kingdom. A stock solution of Nisaplin, a commercial preparation containing nisin, was prepared according to Lauková et al. [25] and used at a dose 10 µg meaning concentration 10 µg/mL. Gallidermin was supplied by Enzo Life Sci. Corporation, Blvd Farmingdale, NY, USA (MW 2069.4), used at a concentration of 0.5 mg/mL (dose 5 µL). Seventeen staphylococcal strains were tested for bacteriocin susceptibility using the agar diffusion method according to De Vuyst et al. [26]. Brain–heart infusion/broth supplemented with 1.5% agar (BHA, Difco, Lawrence, KS, USA) was used for the bottom layer. For the overlay, 0.7% BHA was used containing 200 µL of an 18-h culture of the indicator strain with absorbance (A_600_) of up to 0.800. Dilutions of bacteriocins, in doses of 10 µL and 5 µL (decided on the basis of our previous tests) at 1:1 ratio in phosphate buffer (pH 6.5), were aliquoted onto the surface of soft agar for each tested indicator (staphylococcal) strain. The plates were incubated at 37 °C for 18 h. Clear inhibition zones around each dose of bacteriocins were checked and the antimicrobial activity was expressed in arbitrary units per milliliter (AU/mL), which indicates the reciprocal of the highest twofold dilution of bacteriocin demonstrating complete growth inhibition of the indicator strain, and in the case of nisin and gallidermin also as the minimum inhibitory concentration (MIC, µg). The positive control was the principal indicator strain *Enterococcus avium* EA5 (our isolate from feces of piglets), the growth of which was inhibited by bacteriocin activity of between 25,600–204,800 AU/mL.

## 3. Results and Discussion

### 3.1. Microbial Characterization

The counts of staphylococci in ewe’s milk cheeses were quite high. The total count of colonies grown on baird-parker (BP) agar reached 4.14 ± 2.0 log 10 CFU/g on average, while colonies on MSA agar reached 4.02 ± 2.0 log 10 CFU/g on average. BP agar is mostly used for detection of coagulase-positive staphylococci (CoPS), including *S. aureus*; MSA agar usually serves for determining coagulase-negative staphylococci (CoNS). Šimko and Bartko [27] previously detected staphylococci in ewe’s milk in lower counts (up to 3.0, log 10, CFU/mL). After eliminating identical colonies using MALDI-TOF spectrometry score evaluation, 17 identified strains were taxonomically allotted to the following species: *S. aureus* (5 strains), *S. xylosus* (3 strains), *S. equorum* (1 strain) *S. succinus* (5 strains), and *S. simulans* (3 strains) (Table 1). Variability in staphylococcal species was determined, and five different species were detected. *S. aureus* is a representative of CoPS; the rest of the identified strains species belong among CoNS (Table 1). According to Takashi et al. [5], based on 16S rRNA gene similarity analysis, the staphylococcal species generally fall into 11 clusters. Following this division, staphylococci detected in ewe’s milk lump cheeses can be allotted to three clusters/groups involving five species, *S. aureus* group with five strains of *S. aureus*; *S. simulans* group with three strains of *S. simulans;* and *S. saprophyticus* group involving five strains of *S. succinus*, three strains of *S. xylosus*, and one strain of *S. equorum*. *S. xylosus* SXOS7/2 was evaluated with a score associated with highly probable species identification (2300–3000). Eight strains were measured reaching score corresponding with secure genus identification and/or probable species identification, and another eight strains reached score corresponding with probable genus identification (Table 1). Most parameters tested in the commercial phenotypic strips resulted with the parameters of the reference species strains presented in Bergey‘s Manual of Systematic Bacteriology [28]. Some differences in species strains can appear due to different sources of their isolation.

As mentioned previously, the genus *Staphylococcus* consists of Gram-positive bacteria belonging to phylum Firmicutes, and the family Staphylococcaceae. So far, this genus includes 58 known species with 25 subspecies. In this study, the most commonly detected were representatives of the *S. saprophyticus* group (3 species, 9 strains). Kačániová et al. [29] detected *Staphylococcus aureus* and *Staphylococcus pasteuri* in bryndza, traditional Liptov sheep cheese. Detection of *Staphylococcus equorum* and *Staphylococcus xylosus* in cheeses was mentioned by Even et al. [30]. There is only limited information regarding the microbiota in ewe’s milk or cheeses; mostly only detection is reported but representatives have not been studied in detail.

### 3.2. Antimicrobial Phenotype Testing, Determination of Nuclease, and Hemolysis Activity

The identified staphylococci were mostly susceptible to antibiotics. They were susceptible to 10 out of 14 antibiotics, namely penicillin, erythromycin, tobramycin, streptomycin, ampicillin, cefoxitin, chloramphenicol, azithromycin, vancomycin, and rifampicin. Resistance was only rarely found (Table 2). The tested strains were each resistant only to one or two antibiotics out of the 14 used, except for *S. xylosus* SXOS2/3, which was resistant to three antibiotics—clindamycin, oxacillin, and lincomycin. *S. xylosus* SXOS7/2 was resistant to neomycin. Neomycin resistance was also found in *S. aureus* SAOS2/1. *S. sciuri* Sci OS18/1 was resistant to lincomycin (Table 2). However, the most important finding was that all strains except for strains SXOS2/3 and SAOS2/1 were susceptible to cefoxitin and oxacillin, in which resistance was associated with methicillin [31], meaning that only two strains tested could be methicillin-resistant; however, presence of *mec* gene was not tested there. Susceptibility to antibiotics is one of the properties indicating their non-pathogenic character.

Moreover, these staphylococci showed _γ_- hemolysis; it means they did not form hemolysis, except for the strain *S. aureus* SAOS1/1 forming β- hemolysis (Table 2). This strain also tested positive for DNase, as well as two other *S. aureus* strains SAOS5/2 and SAOS51/3. The rest of the staphylococci were DNase-negative (Table 2). In fact, DNase is a virulence factor which catalyzes the degradation of DNA. However, DNase-positive staphylococcal strains were mostly susceptible to bacteriocins.

### 3.3. Biofilm Testing as Virulence Profile

Using the Congo red agar method to evaluate biofilm, we found only two strains which were positive (*S. aureus* SAOS1/1 and SAOS51/3). The rest of the strains did not form biofilm on Congo red agar (Table 3). However, using quantitative plate assay, 12 strains out of 17 showed low-grade biofilm formation (0.1 ≤ A_570_ < 1, Table 3). Five strains did not form biofilm at all (A_570_ < 0.1). Those two strains of *S. aureus* with positive biofilm formation on Congo red agar were also found to be positive using plate assay (Table 3).

Biofilm formation is assumed to be a factor of pathogenicity especially in pathogenic/spoilage bacteria; biofilm growth protects bacteria against host defenses and the action of antimicrobial agents; therefore, biofilm can be a source of persistent infection [32]. However, the staphylococci in this study were only low-grade biofilm-positive (Table 3). *S. aureus* are often detected as biofilm-positive [33]. In this study, four *S. aureus* strains out of five showed biofilm formation (Table 3). However, biofilm formation was also found in the strains of *S. simulans, S. succinus*, and *S. xylosus*. It seems that biofilm ability was not species-dependent.

*S. aureus* is one of the causative agents of mastitis in dairy herds. It could be speculated that the *S. aureus* detected in our ewe’s milk lump cheeses may have this origin [34], even though our cheeses were made from milk sampled from healthy animals. The major sources of contaminated milk and milk products are usually considered to be improperly cleaned and sanitized farm and factory equipment. It is speculated that contaminants in milk samples may show biofilm formation [6], as shown in the staphylococci from our tested cheeses. Dairy products are very susceptible to contamination by biofilms, and it is challenging to eliminate such microbiota. How can this be done? One possibility is indicated in this study: by using bacteriocins.

### 3.4. Treatment with Bacteriocins

Only three out of five *S. aureus* strains tested were resistant to the enterocins used (Table 3). *S. aureus* SAOS2/1 was susceptible to all enterocins with inhibition activity ranging from 800 up to 12,800 AU/mL, and while growth of *S. aureus* SAOS51/3 was inhibited by treatment of enterocin (ent) M, A/P, and 412, this was associated with lower inhibition activity (100 AU/mL). *S. aureus* strains often show resistance against enterocins under in vitro conditions; however, they can be inhibited under in vivo conditions [24]. In vivo, e.g., fecal CoPS isolated from broiler rabbits were significantly decreased (*p* < 0.05) after application of durancin ED26E/7 and ent M as well. Among the species *S. xylosus*, growth of two strains was inhibited using all enterocins (ents), but SXOS13/4 was inhibited only after treatment with ent A/P and ent 412 (100 AU/mL). Regarding the *S. succinus*, three strains were inhibited with all ents; SciOS17/4 and SciOS8/1 were resistant to ent M only. The situation was similar with *S. simulans* strain SmiOS18/4 (Table 3). The growth of SQOS54 was inhibited. In general, most strains were resistant to ent M but susceptible to other enterocins. The enterocins used in this study are classified as thermo-stable, small peptides with broad antimicrobial effect (involving Gram-negative bacteria) belonging in enterocins classification group II [8,21,22,23,24], which act as pore-forming substances in the cell membrane. Their inhibition activity was reported previously under both in vitro as well as in vivo conditions [14]. In horses, administration of ent M led to mathematical reduction of coliforms, campylobacters (*p* < 0.05), and significant reduction of *Clostridium* spp. (*p* < 0.001). Ents 412, Ent A/P (EK13), Ent M were also effective against various staphylococci isolated from free-living trout [35]. Lauková et al. reported inhibition of *Salmonella enterica* serovar Dusseldorf by ent A in gnotobiotic Japanese quails. There stronger therapeutic than prophylactic effect of ent A (produced by *E. faecium* EK13 = CCM7419) was observed [36]. Significant decrease in CoPS as well as phagocytic activity stimulation was also noted in broiler rabbits after durancin ED26E/7 administration [24].

Nisin and gallidermin are two lantibiotic bacteriocins which are known to inhibit predominantly Gram-positive cocci. The growth of all strains tested, also including those resistant to enterocins, was inhibited with high inhibition activity by both nisin and gallidermin, with a measure inhibition zone ranging from 1600 up to 102,400 AU/mL; MIC was in the range from 0.0625 µg up to 0.005 µg (Table 4). Nisin is a peptide composed of 34 amino acid residues. Gallidermin belongs in the group of polycyclic proteinaceous bacteriocins. They both contain unusual amino acid residues such as lanthionine, β-methyllanthionine or α, β-didehydroamino acids, which are capable of building intramolecular thioether bridges, and so this group of bacteriocins is called lantibiotics [10,37]. The primary site of nisin action against cells is the cytoplasmic membrane; nisin disrupts the membrane by forming pores [37]. The mode of action of gallidermin is similar, as it integrates into the plasma membrane, forming pores and inhibiting cell membrane synthesis [38]. Similarly, as in this study, variable fecal staphylococcal strains of different species isolated from ostriches and pheasants, and producing biogenic amines, were inhibited by gallidermin, reaching inhibition zones in the range from 3200 up to 25,600 AU/mL [39]. All results from previous studies and the present one underline the effective use of enterocins and lantibiotic bacteriocins against contaminating bacteria, representing an opportunity which could be used in further practice. Their beneficial potential against clinical multiresistant staphylococci has already been indicated using aureocin A53 and epidermin [40]. Of course, additional studies are required.

## 4. Conclusions

Seventeen different staphylococci were identified in local Slovak ewe’s milk lump cheeses. They belong in five species and three clusters/groups. These strains were mostly susceptible to antimicrobials and they showed low-grade biofilm ability. In addition, they were susceptible to lantibiotic bacteriocins, nisin, and gallidermin but their growth was mostly inhibited using enterocins. This is a contribution to the knowledge on the microbiota in lump cheeses made from ewes’ milk, and also an indication of how to use bacteriocins for their prevention and/or elimination.

## Figures and Tables

**Table 1 foods-09-01335-t001:** Matrix–assisted laser desorption ionization time-of-flight (MALDI-TOF) spectrometry scores and phenotypization of staphylococci isolated from local Slovak ewes ‘milk lump cheeses.

Strains	Score	Selected Parameters of API STAPH Profile												
		Mal	Lac	Tre	Xyl	Meli	Nit	Pal	VP	Xy	S	Nag	Adh	Urea
SXOS7/2	2.309	+	+	+	−	+	−	±	+	+	−	+	−	+
SXOS13/4	1.708	+	+	+	−	−	±	+	+	−	−	+	−	+
SXOS2/3	1.870	−	−	−	−	−	+	+	±	−	−	−	+	−
SciOS6/3	1.925	+	+	+	−	+	+	+	+	+	−	+	−	+
SciOS17/4	1.986	+	+	+	−	−	±	+	+	+	−	+	−	+
SciOS8/1	1.962	+	+	+	−	−	+	+	+	+	−	+	−	+
SciOS18/1	1.832	+	+	+	−	−	±	+	+	+	−	+	−	+
SciOS5/1	2.032	+	+	+	−	±	−	+	+	+	+	+	−	+
SmiOS17/6	1.703	+	+	+	−	−	+	+	+	−	−	+	−	+
SmiOS14/1	2.000	+	+	+	−	+	+	+	+	+	−	+	−	+
SAOS1/1	2.050	+	+	+	−	−	+	+	+	−	+	+	+	+
SAOS2/1	2.054	+	+	+	−	+	+	+	+	+	+	−	−	−
SAOS6	2.162	+	+	+	+	−	+	+	+	+	+	+	+	−
SAOS5/2	2.185	+	+	+	+	−	+	+	+	+	+	+	+	−
SqOS54	2.000	+	+	+	−	−	+	+	+	+	+	+	±	+

All strains fermented glucose, fructose, mannose (+) and were MDG negative (-). Mal—maltose, Lac—lactose, Tre—trehalose, Xyl—xylitol, Meli—melibiose, Nit—nitrates, Pal—alkaline phosphatase, VP—acetyl-methyl-carbinol production, Raf—raffinose was negative, Xy—-xylanose, S—sucrose, Nag–N-acetyl glucosamine, Adh—arginine dihydrolase, SX—*Staphylococcus xylosus*, Sci—*Staphylococcus sciuri*, Smi—*Staphylococcus simulans*, SA—*Staphylococcus aureus*, Sq—*Staphylococcus equorum*; ±, dubious reaction.

**Table 2 foods-09-01335-t002:** Antimicrobial phenotype profile of detected staphylococci, hemolysis, and DNase test.

Strains	Da	Ox	N	L	H	DNase
SXOS7/2	20	12	R	18	_γ_	ng
SXOS13/4	26	18	18	18	_γ_	ng
SXOS2/3	R	R	20	R	_γ_	ng
SciOS6/3	25	16	20	11	_γ_	ng
SciOS17/4	25	18	21	15	_γ_	ng
SciOS8/1	26	17	23	13	_γ_	ng
SciOS5/1	24	18	22	18	_γ_	ng
SciOS18/1	30	15	18	R	_γ_	ng
SmiOS17/6	22	20	21	18	_γ_	ng
SmiOS14/1	20	18	12	18	_γ_	ng
SmiOS18/4	26	22	16	18	_γ_	ng
SAOS1/1	20	20	23	18	β	+
SAOS2/1	25	R	R	18	_γ_	ng
SAOS6	25	18	18	R	_γ_	ng
SAOS5/2	24	16	21	12	_γ_	+
SAOS51/3	24	19	15	18	_γ_	+
SqOS54	24	23	23	18	_γ_	ng

SX—*Staphylococcus xylosus*, Sci—*Staphylococcus succinus*, Smi—*Staphylococcus simulans*, SA—*Staphylococcus aureus*, Sq—*Staphylococcus equorum*, R—resistant, (No)—inhibition zone size in mm; Da-clindamycin-2 µg, Ox-acillin-1 µg, N-neomycin-, L-lincomycine-2 µg; All strains were susceptible to penicillin (10 IU), erythromycin, azithromycin (15 µg), tobramycin, ampicillin (10 µg), streptomycin, vancomycin, cefoxitin, rifampicin, and chloramphenicol (30 µg); H—hemolysis, γ—hemolysis is negative, β-hemolysis, complete hemolysis, DNase test-negative, ng; +—positive.

**Table 3 foods-09-01335-t003:** Biofilm formation (72 h on Congo red agar, plate assay (PA)) and susceptibility to enterocins.

Strains	72h	PA	M	A/P	412	26E/7
SXOS7/2	ng	0.103 (0.32)	51,200	12,800	25,600	12,800
SXOS13/4	ng	0.106 (0.32)	ng	100	100	ng
SXOS2/3	ng	0.170 (0.41)	6400	6400	800	800
SciOS6/3	ng	0.143 (0.38)	6400	25,600	100	12,800
SciOS17/4	ng	0.137 (0.37)	ng	100	100	100
SciOS8/1	ng	0.084 (0.03)	ng	100	100	200
SciOS5/1	ng	0.073 (0.03)	12,800	25,600	6400	12,800
SciOS18/1	ng	0.089 (0.03)	6400	12,800	25,600	12,800
SmiOS17/6	ng	0.104 (0.32)	12,800	25,600	25,600	6400
SmiOS14/1	ng	0.080 (0.02)	12,800	25,600	12,800	25,600
SmiOS18/4	ng	0.109 (0.33)	ng	100	100	100
SAOS1/1	+	0.167 (0.40)	ng	ng	ng	ng
SAOS2/1	ng	0.190 (0.43)	800	12,800	12,800	6400
SAOS6	ng	0.097 (0.03)	ng	ng	ng	ng
SAOS5/2	ng	0.171 (0.41)	ng	ng	ng	ng
SAOS51/3	+	0.155 (0.39)	100	100	100	ng
SqOS54	ng	0.141 (0.37)	100	100	100	100

SX—*Staphylococcus xylosus*, Sci—*Staphylococcus succinus*, Smi—*Staphylococcus simulans*, SA—*Staphylococcus aureus*, Sq—*Staphylococcus equorum*, 72 h on Congo red agar, PA-plate assay of biofilm (±SD), enterocin M, ent A/P, ent412, durancin ED26E/7; inhibition activity is expressed in arbitrary units per mL (AU/mL).

**Table 4 foods-09-01335-t004:** Susceptibility of staphylococci to nisin and gallidermin, expressed in arbitrary unit per mL and in minimal inhibition concentration (MIC, µg).

Strains	Nis	Nis_MIC_	Gall	Gall_MIC_
SXOS7/2	102,400	0.005	102,400	0.005
SXOS13/4	102,400	0.005	102,400	0.005
SXOS2/3	51,200	0.019	51,200	0.019
SciOS6/3	51,200	0.019	51,200	0.019
SciOS17/4	102,400	0.005	102,400	0.005
SciOS8/1	102,400	0.005	102,400	0.005
SciOS5/1	102,400	0.005	51,200	0.019
SciOS18/1	51,200	0.019	102,400	0.005
SmiOS17/6	51,200	0.019	102,400	0.005
SmiOS14/1	102,400	0.005	51,200	0.019
SmiOS18/4	6400	0.0156	102,400	0.005
SAOS1/1	102,400	0.005	102,400	0.005
SAOS2/1	25,600	0.0039	1600	0.0625
SAOS6	6400	0.0156	51,200	0.019
SAOS5/2	12,800	0.0078	51,200	0.019
SAOS51/3	6400	0.0156	102,400	0.005
SqOS54	6400	0.0156	102,400	0.005

SX—Staphylococcus xylosus, Sci—Staphylococcus succinus, Smi—Staphylococcus simulans, SA—Staphylococcus aureus, Sq—Staphylococcus equorum, nisin, gallidermin.

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
