# Peer review of "Susceptibility to Bacteriocins in Biofilm-Forming, Variable Staphylococci Isolated from Local Slovak Ewes’ Milk Lump Cheeses"

_foods, 2020, doi:10.3390/foods9091335_

Round 1
Reviewer 1 Report
The authors isolated 17 Staphylococcus strains from ewes' milk lump cheeses, which belonged to five species and three groups, and examined their various phenotypes, i.e., resistance to antibiotics and bacteriocins, hemolysis, DNase production, and biofilm formation. Their data showed that, in general, these strains were susceptible to antibiotics and bacteriocins (enterocins, nisin and gallidermin), were negative for hemolysis and DNase, and formed low-grade biofilms. Overall, experiments appear to have been carefully conducted and presented, and the data were appropriately interpreted. However, several issues remain to be addressed to improve the manuscript.
- The current manuscript needs clarification in various places. Please take a look at specific comments.
- Tables should be extensively revised.
(1) Columns are not properly aligned.
(2) Legends are genuinely confusing. It is not always clear which symbol/acronym was explained. Also, some symbols/acronyms were absent in the legends while symbols/acronyms which were not shown in the table were sometimes explained in the legend.
- The strains isolated in this study were divided into three groups. However, it is not clear how three clusters/groups were determined. Was 16S rRNA gene sequenced?
- References are not shown for many statements. Show the references.
- Cite tables more frequently in appropriate places.
- "Table 1" should be called "Supplemental Table 1".
- In the Materials and Methods section, separate the product acronym, manufacturer information and its concentration with a semi-colon throughout the manuscript. For instance, in line 85, change "MSA Becton and Dickinson" to "MSA; Becton and Dickinson". Also, geographical location of some manufacturers are not fully shown. Provide the complete information.
Specific comments:
Abstract
Lines 16 and 17: Remove "Variability ... identified" since it is redundant.
Line 19: Change "to 10 out of 14" to "10 out of 14".
Line 20: Revise "hemolysis negative-gamma hemolysis)" to improve clarity.
Lines 21 and 22: "This strain ... DNase negative." can be further condensed.
Line 24: Merge "Five strains ... biofilm (A570 < 0.1)." with the prior sentence.
Line 25: Add more information on "those resistant to Enterocins". Change "inhibited using" to "inhibited by".
Line 26: Change "reaching" to "resulting in". Show the full name for "AU".
Lines 26-28: Revise the sentence to improve clarity.
Introduction
Line 32: Change "among which" to "to which".
Line 33: Change "calcium, zinc, phosphorous" to "calcium, zinc, and phosphorus".
Line 34: Change "allergic for example to" to "allergic, for example, to".
Line 37: Change "such cheese consumption" to "the consumption of such cheese". Change "persons" to "people".
Lines 40-42: Show references.
Lines 43 and 44: Provide references.
Line 47: Clarify "treatening of raw material".
Line 53: Change "grain sized" to "grain-sized".
Line 54: Change "slight acidic" to "slightly acidic".
Line 59: Change "in group of bacteria" to "in a group of bacteria".
Line 64: Change "Special group" to "A special group".
Line 66: Clarify "spread antimicrobial effect".
Lines 68 and 69: Revise "For consumers and producers as well is important to know".
Line 70: Revise "influence their character and consumption". Add "However," in front of "Only limited ... ".
Lines 73 and 74: Revise "some of their ... gallidermin" to improve clarity.
Materials and Methods
Line 85: Add a comma after "tellurite".
Lines 85: Double-check "Becton and Dickinson".
Line 86: Change "USA, as recommended with ISO)" to "USA) as recommended with ISO".
Line 87: Change "colony forming unit per gram (CFU/g, log 10)" to "log 10 colony forming unit per gram (log10 CFU/g).
Lines 90 and 91: What is "a Microflex ... (Germany)"? I am perplexed here since MALDI-TOF machine was already mentioned.
Lines 91 and 92: "This method ... microbiology" does not seem to be necessary. Remove it.
Lines 94-96: Revise the sentence to improve clarity.
Line 97: Change "colonies evaluated with" to "colonies with".
Line 102: Remove "using phenotypization-". Change "strips; this means" to "strips, which are".
Lines 104 and 105: Re-write "Each microtube ... medium" to improve clarity.
Line 105: Change "18-24 hours incubation" to "18-24 hour incubation".
Line 108: Replace the comma in "ATCC 27848, American" to a semi-colon.
Line 109: Change "involved are:" to "involved are as follows:".
Line 114: Move "[16]" to the end of a sentence.
Line 117: Change "neomycin 10 μg),," to "neomycin (10 μg),".
Line 118: Change "cefoxitin" to "and cefoxitin".
Line 123: Change "(Oxoid, USA)" to "(Oxoid)".
Lines 123-125: "The production ... medium" does not seem to be necessary.
Line 129: Show the manufacturer of sheep blood.
Line 132: Change "Slime production" to "slime production".
Line 133: Change "belongs among" to "belongs to".
Line 134: Use a colon instead of the semi-colon in "methods; the qualitative".
Line 136: Separate the manufacturer information and concentration with a semi-colon.
Line 137: Show the manufacturer information for sucrose and pure agar.
Line 140: Clarify "Double testing was performed".
Lines 141 and 142: Revise "Biofilm formation ... identified strains" to improve clarity.
Lines 144: Change "no 1" to "no. 1".
Line 145: Change "brain- heart infusion (BHI)" to "BHI".
Lines 164 and 165: What does "23" represent in "horse strain, 23"?
Lines 166 and 167: Revise "Mareková et al. [21, 22], and 166 [23, 13, 14]".
Line 167: Start a new sentence from "nisin was supplied ... ".
Line 168: Show the manufacturer of Nisaplin.
Line 170: Clarify "(dose)".
Line 171: Revise "and dose of 5 uL".
Line 172: Change "bacteriocins susceptibility" to "bacteriocin susceptibility".
Lines 172 and 173: Change "Brain heart infusion" to "BHI". Is "BHIA" the same as "brain heart agar" in line 143? If so, use a consistent term.
Line 174: Change "enriched with" to "containing".
Line 176: Add a comma after "(pH 6.5)".
Lines 182 and 384: Change "MIC-ug" to "MIC, ug".
Lines 183 and 184: Condense "our isolate from ... Slovakia". I do not think that the institute name is necessary here.
Results and Discussion
Line 188: "Baird-Parker agar" was previously mentioned. Show the acronym when the full name first appears and use the acronym consistently afterwards.
Line 189: Change "CFU/g (log10)" to "log10 CFU/g".
Lines 189 and 190: Move "The counts of … high" to the very beginning of the paragraph.
Line 193: Change ", CFU/mL, log 10" to " log10 CFU/mL".
Line 195: Change "Staphylococcus aureus" to "S. aureus".
Line 196: Change "(3 strains, Table 1)" to "(3 strains) (Table 1)".
Lines 196 and 197: This is redundant; remove it.
Line 198: Change "genes similarity" to "gene similarity".
Lines 200-202: Italicize the species names before "group".
Lines 203-208: It is hard to understand this part. Revise it to improve clarity.
Line 210: Change "Firmicutes, and the family" to "Firmicutes and the family".
Lines 211 and 212: Italicize "S. saprophyticus" in "S. saprophyticus group".
Lines 212 and 213: Remove "the species" in front of "S. aureus" and "S. equorum".
Lines 220 and 221: Change "Only rare resistance was found;" to "Resistance was only rarely found.".
Lines 223-225: This sentence can be placed in line 221.
Line 226: Move "except for strains SXOS2/3 and SAOS2/1" behind "all strains" in line 225.
Lines 227 and 228: Change "methicillin resistant" to "methicillin-resistant".
Line 228: Were mec genes not tested or not found in strains SXOS2/3 and SAOS2/1?
Line 230: Revise "gamma- hemolysis meaning hemolysis-negative".
Line 232: Change "DNase negative" to "DNase-negative".
Line 234: Clarify "these strains".
Lines 281 and 282: "Ten strains … used" is redundant; remove it.
Line 288: Change "low-grade biofilm positive" to "low-grade biofilm-positive". Revise "detected as biofilm-forming".
Line 289: Remove "identified".
Line 290: Change "strains S. simulans" to "strains of S. simulans".
Line 291: "But they were ... bacteriocins" does not seem to be relevant here. Start a new paragraph from "S. aureus".
Line 345: "growth of S. aureus SAOS51/3 was inhibited" by what?
Line 346: Show the numbers inside the parentheses.
Lines 347-349: Add a comma after "For example". In fact, is this entire statement relevant here?
Line 351: What do "Ents" and "Ent M" mean?
Lines 352 and 353: Clarify "The situation ... inhibited".
Lines 355 and 379: Change "Enterocins" to "enterocins".
Line 356: Add a comma before "which".
Line 357: Change "both, in vitro as well as" to "both in vitro as well as".
Line 358: What does "mathematical reduction" mean?
Line 360: Move "[36]" to the end of a sentence.
Lines 361 and 362: Revise "In these birds ... observed".
Line 363: Change "were also noted" to "was also noted".
Line 365: Add a comma before "which".
Line 369: Change "aminoacid" to "amino acid". Change "belongs in" to "belongs to".
Line 371: Add a comma before "which".
Line 377: Add a comma after "amines". Change "inhibited with gallidermin" to "inhibited by gallidermin".
Line 383: Change "Arbitrary" to "arbitrary".
Conclusion
Lines 427 and 428: Change "and their growth" to "but their growth". Change "using Enterocins" to "by enterocins".
Lines 428 and 429: Revise the sentence.
Author Response
Cover letter-Foods
I checked according to test similarity and on recommendation, the text is changed. Similarity with L. Belej etc. (16%) is not correct because Belej, Capla are editors of Proceedings from conference and contribution is mine or colleague of our team, it means from our lab partially referred results at Conference; it means only in Proceedings. I dont know why the system involve it like this. It is our work.
I hope that the other things were revised and involved or explained. I tried to improve the table 1 I also indicated that our strains were divided according to takashi, so now i tis more clear.
Phenotypization Table -1 was kept because one reviewer gives question i fit is necessary but second one has not this request.

Reviewer 2 Report
The submitted manuscript is about the isolation and identification of Staphylococci strains from local Slovak ewes‘ milk lump cheeses and some properties such as susceptibility to antibiotics and some bacteriocins and also biofilm production.
Staphylococci is a major food safety issue especially concerning ewes‘ milk and products made from ewes‘ milk. So any information regarding this pathogen is of great importance.
The Introduction and the description of the state of the art are clearly articulated.
Experiments are carefully designed and performed. In line 87 is referred by the authors that “…different colonies were picked up,…”. It would better if the authors can specify the exact methodology used for the selection of colonies, for example the number of colonies, in order to facilitate the reproducibility of methods used in the present study.
The Results are sufficiently analyzed and interpreted. Furthermore, authors’ conclusions are supported by the results and are related to the literature. Although I consider myself an experienced reader in scientific manuscripts, it was very difficult to follow the important information provided by Tables 3 and 4. I suggest to the authors to modify Tables 3 and 4 in a more comprehensible way for the reader.
The manuscript is well written concerning the English grammar, style and syntax. Some indicative corrections:
- Line 39: Please change “undefinied” to “undefined”
- Line 441: Please change “availabre” to “available”
- Lines 480-486: Please change references 15 and 16 according to instructions for authors
Author Response
Detail revisions and responses to Reviewer 2
Thnk so much reviewer for her/his kindness to check the manuscript and gives us chance to improve it. Everything done or revised is in red colour directly in the text.
Line 87, 80 colonies
I tried a little bit modify Tables 3 and 4, but it is very hard when to keep rules for tables managing in Foods journal, I mean a style. So, I tried to change or to clear legenda.
Line 39...it was done, Line 441, it was changed...available;
Lines 480-486, revised.
